



# Low methane emissions from a boreal wetland constructed on oil sand mine tailings

M. Graham Clark[1,2], Elyn R. Humphreys[1], and Sean K. Carey[2]

[1]Department of Geography and Environmental Studies, Carleton University, Ottawa, Ontario, K1S 5B6
[2]School of Geography and Earth Sciences, McMaster University, Hamilton, Ontario, L8S 4L8

*Correspondence to*: M. G. Clark (dr.mg.clark@gmail.com)

**Abstract.** A 58 hectare mixed upland and lowland boreal plains watershed called the Sandhill Fen Watershed was constructed between 2008 and 2012. In the years following wetting in 2013, methane emissions were measured using manual, static, non-steady state chambers. The presence of vegetation with aerenchymous tissues and saturated soils were important factors influencing the spatial variability of methane emissions across the constructed watershed. Nevertheless, median methane emissions were equal to or less than 0.51 mg $CH_4$ $m^{-2}$ $h^{-1}$ even from the saturated organic soils in the lowlands. Although overall methane emissions remained low, observations of methane ebullition increased over the three study years. As a ratio to the total number of measurements, the number of ebullition events increased from 10 % in 2013 to 21 % and 27 % in 2014 and 2015, respectively at the plots with saturated soils. Increasing metal ion availability and decreasing sulphur availability was measured using buried ion exchange resins at both seasonal and annual timescales potentially as a result of microbial reduction of these ions. Methane fluxes significantly correlated with the leading Principal Component of ammonium, iron, manganese and sulphur availability (r = 0.31, p < 0.001). These results suggest that an abundance of alternative electron acceptors may be limiting methanogenesis at this time.





## 1 Introduction

The boreal biome stores large quantities of soil carbon (C) due to the high density of peatlands with some estimates as high as 165 kg C $m^{-2}$ (Beilman et al., 2008). A legacy of open pit oil sands mining in northern Alberta, Canada will be that almost 5000 $km^2$ of the boreal landscape (Alberta Government, 2017) will require reclamation to restore it to

the "equivalent capability" of the pre-mining ecosystems (Enviornmental Protection and Enhancement Act 2017). These engineered landscapes will be important in determining the long-term C footprint of the region, potentially helping to offset ecological losses and industrial emissions. Rooney et al. (2012) used land cover classification and reclamation plans to estimate a potential net loss of 11.4 – 47.3 Tg of the stored soil C from the pre-mining landscape and a reduction in $CO_2$ sequestration capacity of the new constructed landscape by 5.7 – 7.2 Gg C $year^{-1}$. This loss is

expected to occur as a result of both the decomposition of peat when it is used in the reclamation of uplands and from the diminished C sequestration capacity of engineered landscapes. In particular, Rooney et al. (2012) predicted that an insufficient proportion of the disturbed area will be replaced with the C sequestration capacity of the original boreal wetlands. However, wetland reclamation in the region is still quite novel, and little is known about the surface-atmosphere exchange of C in these newly engineered wetlands (Nwaishi et al. 2015; Clark et al. 2019).

Undisturbed peatlands are long-term $CO_2$ sinks, accumulating C as peat over millennia (Rydin and Jeglum, 2006, p 250; Vasander and Kettunen, 2006). Yet, on annual timescales the C cycling within peatlands is highly variable and only partially understood, particularly with regards to methane ($CH_4$) emissions (Vasander and Kettunen, 2006). Wetland $CH_4$ emissions are thought to be the single greatest driver of inter-annual variability in atmospheric $CH_4$ concentration (Bousquet et al., 2006) and globally wetlands are the largest non-anthropogenic source of $CH_4$ to the

atmosphere (Kirschke et al., 2013). Atmospheric carbon is relevant for it's role in the radiative forcing of the troposphere and over centennial timescales $CH_4$ has 20 times the warming potential as $CO_2$ (IPCC, 2007), but with a half-life of 25 years impacts to radiative forcing is much larger on decadal timescales. Therefore, $CH_4$ production and release is an important land-atmosphere climate change feedback process (Kirschke et al. 2013). Since methanogenesis is only favoured at the lowest reduction/oxidation (REDOX) potentials known to support life, the high rates of $CH_4$

emission and low rates of C turnover in peatlands are linked to a scarcity of inorganic electron acceptors in anoxic, saturated soils (Han-Schofl et al., 2011; Peters and Conrad, 1995; Thomas et al., 2009). Around the time that Rooney et al. (2012) estimated the C balance for the post-mining landscapes of Alberta, the first two large-scale wetlands were being constructed in the Fort McMurray region. These wetlands were designed and constructed to test theories on how to return the peat forming and C capturing capacities to the mined landscape (Wytrykush et al., 2012). A research gap

exists, however, between the modelled predictions of C cycling by Rooney et al. (2012) and the actual C cycling of constructed wetlands. Clark et al. (2019) discussed the increasing $CO_2$ sink strength over the first three years of one of these wetlands but did not address emissions of $CH_4$, which can contribute substantially towards the long-term C balance of some ecosystems (Vasander and Kettunen, 2006). The aim of this study is to evaluate $CH_4$ emissions from this constructed ecosystem and identify the factors which influence their temporal and spatial variability.

Methane emissions from wetlands are highly variable in space and time (Moore et al., 1998). Four key wetland characteristics are typically linked to the temporal and spatial variability of $CH_4$ flux in peatlands: water table position,



REDOX, vegetation composition, and temperature (Kim et al., 2012; Limpens et al., 2008; Turetsky et al., 2014; Vasander and Kettunen, 2006). Approximately 60 to 90 % of the $CH_4$ produced in peatlands is re-oxidized before it reaches the atmosphere (Le Mer and Roger, 2001). Emissions tend to be higher in areas with a high water table as this limits the potential for $CH_4$ oxidation (Moore et al., 2011) while deep rooting vegetation with aerenchymatous tissues (Couwenberg, 2009; Kludze et al., 1993) allow $CH_4$ to bypass the aerated surface peat where aerobic methanotrophic communities reside. Aerenchymatous tissues can also transport oxygen into the rhizosphere which can increase REDOX potentials and create unfavourable niches for anaerobic microbial communities (Blossfeld et al. 2011). Rhizospheric microbial communities readily metabolize the labile organic compounds exuded by roots such that plant net primary productivity has positively correlated with $CH_4$ production (Dacey et al., 1994; Megonigal and Schlesinger, 1997; Vann and Megonigal, 2003; e.g. Whiting and Chanton, 1993; Ziska et al., 1998). Numerous studies have also shown $CH_4$ emissions increase in warmer soils (Bubier et al., 2005; Bubier and Moore, 1994; Roulet et al., 1992; Whalen, 2005; Whiting and Chanton, 1993) although this effect may be masked when water table or other environmental factors co-vary (Moore et al., 2011).

Although constructed peatlands are relatively novel, restoration of peatlands has occurred for decades by rewetting drained peat, often by blocking the ditches originally used to drain them (Rochefort and Lode, 2006). In a comprehensive review of the literature, the IPCC suggested that $CH_4$ emission rates from rewetted boreal sites with rich organic soils can be estimated as 2.1 mg $CH_4$ m$^{-2}$ h$^{-1}$ but encourage site specific study because the range in the published literature is very large and not normally distributed (Blain, et al., 2014). A follow up review of the literature added additional evidence that $CH_4$ emissions were 1.7 to 22.6 times greater for rewetted vs. drained boreal organic soils (Wilson et al., 2016). The data examined by Wilson et al. (2016) was extrapolated using models of average area (to include multiple surface types) and annual rates, to conform with IPCC reporting standards. Studies which directly compare rewetted and drained environments show that rewetted boreal (temperate) organic soils have an average of 23 (64) times the $CH_4$ emissions of their drained counterparts (supplementary Table S1). However these differences in $CH_4$ emissions are highly variable and some studies shown a decrease in $CH_4$ emissions after wetting (Christen et al., 2016; Juottonen et al., 2012; Urbanová et al., 2012; Waddington and Day, 2007).

To determine the $CH_4$ emissions of a closure watershed in the bitumen mining region of northern Alberta and how these emissions compare to those from the restored peatlands listed above, this paper presents three years (2013-2015) of $CH_4$ emission measurements from the Sandhill Fen Watershed (SFW), one of the first wetland complexes constructed in the Athabasca oil sands region (AOSR). Once water was added to the SFW, approximately 17 ha of the 58 ha construction had permanently saturated peat soils with little horizontal or vertical water flow (Nicholls et al., 2016) and thriving plant communities including species with aerenchyma (e.g. *Carex* spp.) (Vitt et al., 2016). We hypothesized that these areas would be a significant source of $CH_4$ and an important component of the C balance of the SFW. Results of this research will provide important information on the greenhouse gas and C budget for reclamation ecosystems in the AOSR and help guide future strategies for ecosystem design and carbon management.



## 2 Study Site

The SFW (57.0403 N, 111.5890 W) was designed to reclaim sand-capped soft tailings with the conditions needed to promote the long-term development of a peat forming boreal plains ecosystem (Wytrykush et al., 2012). Details of the SFW construction were described by Wytrykush et al. (2012) and others (Biagi et al., 2019; Nicholls et al., 2016;

Oswald and Carey, 2016; Vitt et al., 2016). Clark et al. (2019) described the three main topographic features of the SFW as a lowland region (the wetland), a midland region (drained but moist organic soils), and upland region lying 2 to 6 m above the wetland with well drained sandy soils and two experimental perched wetland sites with moist organic soils (Figure 1). The sand of the lowland, midland and perched wetland regions were covered with 0.5 to 1 m peat coarsely mixed with some underlying mineral soil via excavation of nearby peatlands before mining. A goal of the

SFW design was to limit vertical transport of tailings and process-affected waters (i.e. water with some component of industrial wastewater) to the surface (Wytrykush et al., 2012). To achieve this, fine sediments with 26.4 – 46.4 % clay (based on texture analysis) were placed in the lowland regions of the SFW to help minimize vertical transport of process affected waters (known to contain salts and naphthenic acids) from the the tailings below.

Initially, fresh water was pumped into the containment pond in the west end of the SFW in 2013 (Figure 1). This

water flowed through a leaky gravel dam into the SFW to initially saturate the lowland region. A downstream pump was used to remove water from an outlet V-notched weir (Figure 1). Except for a few hours of operation, the pumps were not used in 2014 or 2015. Four underdrains, which run along the central area of the lowland region, were placed to limit the upwelling of the salt-rich process water. The underdrains were not operated after the beginning of the 2014 growing season.

After placement and before wetting, the top 0.25 m of the donor peat/mineral material in the midland and lowland had a mean ($\pm$ SD) C to nitrogen (N) ratio of 22.7 ($\pm$ 3.4), N content was 0.98 ($\pm$ 0.36) % by dry weight, Ca and Na concentrations were 786.1 ($\pm$ 409.4) and 184.2 ($\pm$ 79.0) mg kg$^{-1}$, respectively and electrical conductivity was 1980 ($\pm$ 600) $\mu$S cm$^{-1}$ (n = 107). The concentration of Fe was 8028 ($\pm$ 2712, n=13) mg kg$^{-1}$ and SO$_4^{2-}$ concentration was high, with a mean of 914.0 ($\pm$ 425.5, n=107) mg of S kg$^{-1}$. Total S concentration by dry weight was also high at 1.00 % ($\pm$ 0.50 %, n=12) determined by

oxidation in an induction furnace then quantified through infrared mass spectroscopy with a LECO IR (LECO Corporation, Saint Joseph, MI).

In the third growing season (2015) since the ecosystem was seeded/planted (see Vitt et al., 2016 for more details), there were four distinct plant communities in the lowland roughly distributed along a soil moisture and water depth gradient (Vitt et al., 2016). The areas with standing water (Figure 1) were dominated by *Carex aquatilis* Wahlenb*., Typha latifolia* L. and to a

lesser extent *Carex utriculata* Stokes, while areas with the water table near the surface contained the most species associated with peatlands including sedges and bryophytes [*Vitt et al.*, 2016]. The drier areas contained the largest percentage of weedy species, or species not associated with peatlands (Vitt et al., 2016). The midland and organic soil regions of the upland was populated by planted trees (*Populus tremulodies* Michx., *Betula papyearifera* Marshall, *Picea glauca* Monech, *Picea mariana* Mill., *Larix laricina* Du Roi, and *Pinus banksiana* Lamb.) and local grasses (e.g. *Hordeum jubatum* L.) that had

naturally colonized the region. Vegetation cover throughout the whole watershed increased over time; peak season leaf area index increased from 1.5 $\pm$ 0.6 to 2.2 $\pm$ 1.1 over the three years as measured at 15 plots (5 midland, 10 lowland) using a plant canopy analyzer (LAI-2200, LI-COR Inc., Lincoln, NE).



## 3 Materials and methods

This study used plot-scale measurements of $CH_4$ fluxes over three years to assess the temporal and spatial variations
in $CH_4$ emissions and the factors that influence their variability. Fluxes were measured using non-steady state static
chambers following methods described by Wilson and Humphreys (2010). Plots were established over the three years
of study to sample the dominant landforms and built features of the SFW. In 2013, 15 plots were established across
the lowland and midland of the SFW. In 2014, the number of plots was increased to 21 by including plots on the
organic soils in the perched wetlands (upland region) and in 2015 was increased to 29 by adding more plots in the
lowland (Figure 1). The 2013 plots were chosen before the pumps were activated and were selected to capture the
surface moisture and vegetation heterogeneity while maintaining multiple plots on similar landforms. The midland
plots were chosen to capture differences in the vegetative cover. Two plots were placed on sandy soils, one directly
on exposed tailings sand and the other on the salvaged mineral soil mixture. In 2014, the new collars were placed on
the experimental wetland sites that were depressions built into the upland hills.  Note that these upland plots remained
moist but well drained for the duration of the measurements. The eight new collars in the lowland added in 2015 were
placed along a moisture gradient transect also used for REDOX monitoring (described below).

The midland plots had water table depths over 0.5 m below the surface and all upland plots had water table depths
exceeding 1 m. For the remainder of this paper, midland and upland plots were combined into one group called
'Upland' due to their similar soil, water table depth and vegetation characteristics.  The lowland plots were grouped
according to whether there was standing water at the time of measurement. The 'Saturated' group contained plots with
standing water with all remaining plots in the 'Unsaturated' group.  Water table depth was not monitored at each plot
but soil moisture was monitored using probes described below.

Each plot included a pair of 0.19 m tall collars that were inserted until nearly flush with the soil surface to minimize
changes to the microclimate as a result of the collar (Parkin and Venterea, 2010). Each pair consisted of one collar
that was maintained free of vegetation by clipping and the other was left undisturbed (Figure 2). An inevitable
limitation of the undisturbed collar was that any vegetation larger than the flux chambers (0.4 m tall, 0.03 $m^3$) was
trimmed to fit within the chamber. This trimming had the largest effect in the lowland, where some *Typha* died once
trimmed. The collars had a surface area of 0.07 $m^2$ and were made of SDR35 12" PVC sewer pipe with a groove cut
into the top edge where an acrylic chamber was placed during measurement. The chambers were constructed of acrylic
and were covered in opaque black plastic to reduce heating within the chamber and eliminate photosynthetic uptake
of $CO_2$ for a respiration analysis discussed by Clark et al. (2019). A seal between the chamber and collar was made
by filling the groove with water.  A small coiled vent tube on the top of the chamber maintained equal pressure with
the surroundings (Hutchinson and Livingston, 2001). During a measurement, the air inside the chamber was mixed
by pumping a 60 mL syringe connected to the chamber sampling line and then pulling 24 mL of air from the chamber
volume at 0, 5, 10, 15, and 20 minute intervals. The sampled air was injected into a 12 mL evacuated vial containing
a small amount of magnesium perchlorate to remove any water vapour from the air sample. Flux measurements were
made between late May and early August in all three years, referred to hereafter as the growing season. The air samples
were transported to Carleton University where $CO_2$ and $CH_4$ concentrations were measured on a gas chromatograph



(CP 3800, Varian, CA) within a few months of sampling. The operational details of the gas chromatograph (GC) are

described in Wilson and Humphreys (2010).

During the initial collar installation, five thermocouples were buried to a depth of 2, 5, 10, 20, and 50 cm at each plot between the two collars. At the time chamber flux measurements were made, soil temperatures were recorded from the buried thermocouples. A portable soil water sensor that integrated over the upper 0.2 m of soil (Hydrosense, Campbell Scientific Inc., Utah, USA) was used to measure volumetric water content of the soil at three locations

surrounding each of the collars. When there was standing water over 0.05 m deep, no manual soil water measurements were made. In those cases, the soil moisture was estimated at 87 % (an estimate of saturated conditions based on an average bulk density of 0.28 g cm$^{-3}$).

Fluxes in units of mg C-CH$_4$ m$^{-2}$ h$^{-1}$ were calculated from the linear rate of change in CH$_4$ mixing ratios (nmol CH$_4$ mol$^{-1}$ air), the molar density of the air in the chamber (mol air m$^{-3}$), the chamber volume (m$^3$) and area (m$^2$) and the

molecular weight of carbon (mg C nmol$^{-1}$ CH$_4$). To determine air density, barometric pressure was recorded at a nearby micrometeorological station (Figure 1) and chamber temperature was estimated using the 0.02 m thermocouple at each plot. The chamber volume was adjusted for the different collar heights and depth of standing water. All calculations and statistical analyses were carried out with MATLAB 2015 (Mathworks Inc., Massachusetts, USA).

The $R^2$ coefficient from the liner regression of CH$_4$ concentration over time is not suitable as a quality control metric

by itself (Lai et al., 2012). For example, when the fluxes approach zero, the slope also approaches zero and small variability in gas concentrations results in low $R^2$ values. In total, 16 % of the fluxes had an $R^2$ over 0.9 and 28 % over 0.8. Instead of the typical $R^2$ filtering (Lai et al., 2012), quality control of the data was done by a visual inspection of each time series used to calculate the fluxes. Any flux measurement which had a distinctly non-monotonic or non-linear trend due to individual data point anomalies were removed from further analysis (6.4 %, 12.3 %, 11.2 % of the

flux measurements in 2013, 2014 and 2014, respectively). These fluxes occurred when the concentration time series included 'dropouts' (sudden non-linear decrease in concentration) or 'spikes' (sudden non-linear jump in concentration) and represented fluxes from a leaking chamber or an ebullition event (Tokida et al., 2007). Any time series with a 'spike' at the beginning or during the measurements was flagged as an ebullition event, and the calculated flux was discarded from further analysis.

In 2013 REDOX potential was measured every 15 min with nine HYPNOS III rods (Vorenhout et al., 2011) inserted into the soil every 2 m alongside the 20 m transect of new plots to evaluate the impacts of a moisture gradient on CH$_4$ emissions (Figure 1). The HYPNOS rods were equipped with four platinum probes and thermistor temperature sensors, of which the sensors 0.2 and 0.4 m below the surface were used in this study. The reference was a pH probe with a 0.1 M KCl standard buried below the water table near the middle of the transect. The probes were connected

to two HYPNOS dataloggers and the standardised REDOX potentials (n = 36) were calculated as the sum of the measured potential and the reference potential. No pH correction was applied to probe measurements because pore water remained circumneutral and stable throughout the season. During summer 2015, the mean pH was 7.1 ± 0.4 (± SD) as determined using weekly measurements at a nearby pore water sampling well that integrates the water from ~1 m depth to the surface. Early spring (May 20$^{th}$) had the largest discrepancy between the well and surface water



measurement locations, with the surface water at a pH of 7.6 and the well water at 6.2. By early July the two sampling

locations had almost converged at neutral, with a slightly higher pH in the surface water (~7.4 vs 6.9).

Within each plot, three replicate sets of Plant Root Simulator (PRS) probes (Western Ag., Saskatoon, Canada) were

buried at a depth of 0.1 m (shallow) and 0.2 m (deep) outside the collars. PRS probes are ion exchange membranes

designed to mimic in-situ soil-root exchange of nutrients and other ions in a non-destructive manner (Qian et al., 2008;

Qian and Schoenau, 2005).  PRS probes provide a time-integrated representation of the soil nutrient availability/net

adsorption rates in units of ion mass per membrane area per burial time.  These values differ from the more common

point-in-time soil extraction measurements (typically element mass per mass of dry soil) although studies show good

correspondence between methods for N, P, K, and S (Harrison and Maynard, 2014; Qian et al., 1992). The probes

were buried for 28 days, (~ one month) with three consecutive burial periods monitored each year starting on day 149,

and 146 for 2013, 2014, and 2015, respectively. For simplicity, since the midpoint of each of these three one-

month burial periods corresponds roughly to the start of June, July and August, results from each period were referred

to as the result from those months (i.e. the July ion adsorption rate refers to the moles of ions absorbed by the PRS

probes period between 174th and 202nd day of the year). An example plot with collars, PRS probes and thermocouples

is shown in Figure 2.

Temporal trends in the PRS probe ion data were assessed using Spearman rank correlations using three different

temporal groupings of the data. First, the data were binned into categories 1 to 9, representing each of the three months

across three years of measurements (for example, June of the third year was category 7). Second, the data were binned

into three categories corresponding to the three months of measurements each year, regardless of year, to ignore the

inter-annual trends. Finally, the data were binned by year, regardless of the collection month, to ignore the

monthly/seasonal trends. This allowed some quantification of the seasonal trends relative to the inter-annual trends in

all three groups (upland and the saturated and unsaturated lowland groups).

To relate $CH_4$ fluxes, REDOX and PRS probe results, a principal component analysis (PCA) was performed on the

standardized measurements (z-scores) of net ion adsorption rates from the PRS probes buried at 0.2 m depth in the

nine plots with REDOX probes. The availability of alternative electron acceptors including $NH_4^+$, $Mn^{4+}$, $Fe^{3+}$, and

$SO_4^{2-}$ are linked to varying soil REDOX potential through their participation in microbially mediated REDOX

reactions. It should be noted that although PRS probes adsorb all mobile forms of S, most are expected to be $SO_4^{2-}$ (Li

et al., 2001). The leading two principal components were correlated to the average 0.2 m REDOX measurement and

the average natural logarithm transformed $CH_4$ flux (to account for the positive skew in the $CH_4$ flux data) for the

same burial period. Methane fluxes were increased by a common absolute value of the minimum flux observed in the

study (-0.10 mg C-$CH_4$ m$^{-2}$ h$^{-1}$) to permit the log transformation. Pearson correlation was also used to assess any linear

relationships between the transformed $CH_4$ flux and soil moisture and 0.02 m soil temperature. Individual Pearson

correlations were calculated between the mean transformed $CH_4$ flux and the net ion adsorption rates at the two burial

depths for a burial period.

The effects of vegetation and standing water on $CH_4$ fluxes were evaluated using a linear mixed effect

(LME) model:



$$Methane\ Flux_{ij} \ = \ \alpha_{ij} \ + \ \zeta_{0j} \ + \ \beta_1 Vegetated_{ij} + \beta_2 Saturated_{ij} + \epsilon_{ij} \qquad [1]$$

where *Vegetated* and *Saturated* were binary vectors indicating if the measurement came from a vegetated plot, or a saturated plot, respectively. *i* and *j* subscripts represented measurements from each sampling day and site respectively. β was the slope of the fixed effect, ε was the residual, α was the intercept, and ζ was the random effect from the repeated measures occurring at each site.

An independent but similar LME model was constructed to detect significant effects of soil depth and standing water on PRS net ion adsorption rates and REDOX potentials; the *Vegetated* covariate was replaced with the binary vector *deep*, which represented the relative depth of the REDOX or PRS measurement (i.e. deep or shallow):

$$PRS\ ion/REDOX\ potential_{ij} \ = \ \alpha_{ij} \ + \ \zeta_{0j} \ + \ \beta_1 Deep_{ij} + \beta_2 Saturated_{ij} + \epsilon_{ij} \qquad [2]$$

## 4 Results

The three growing seasons (1 May through 31 September) were warmer and wetter than the long-term average. The average air temperatures recorded at 3 m above the lowland were 16.4, 15.3, and 15.3ºC and total rainfall was 375.1, 299.1, and 231.3 mm for the 2013, 2014, and 2015 growing seasons, respectively. The 1981-2010 climate normal for this period was 13.3ºC and 211.1 mm at a nearby weather station (Fort McMurray Airport; 48 km from study area; Environment and Climate Change Canada, 2016). Over the three-year study period, the upland plot soils were drier and slightly warmer than the lowland plots with an average volumetric soil moisture of 34.7 % compared to the 57.5 % in the lowland (Table 1). There were only small differences in growing season 0.02 m soil temperatures at the plot-level (Table 1). Near the climate monitoring station in the centre of the lowland the EC was measured to be $1171 \pm 269$, $2109 \pm 306$, and $2163 \pm 248$ µS cm$^{-1}$ in 2013, 2014 and 2015, respectively where the typical electrical conductivity of boreal wetland pore waters ranges from 400 to 2770 µS cm$^{-1}$ (Trites and Bayley, 2009).

### 4.1 Spatial and temporal CH$_4$ and ion relationships in the SFW

During each growing season, CH$_4$ emissions were generally very low, with median values less than or equal to 0.04 mg C-CH$_4$ m$^{-2}$ h$^{-1}$ for all plot groups (Table 1). The greatest median CH$_4$ emissions were from the saturated plots (all located in the lowland) in July 2015 with median emissions reaching 0.51 mg C-CH$_4$ m$^{-2}$ h$^{-1}$(Figure 3).

The proportion of measurements where substantive CH$_4$ emissions were detected increased for the saturated plots over the study period (Table 1). These measurements were subjectively defined as flagged ebullition events or when fluxes were greater than 0.5 mg C-CH$_4$ m$^{-2}$ h$^{-1}$. This value is equivalent to ten times the maximum CH$_4$ uptake rate observed at this site, which is also in the upper range of average uptake rates observed in grassland and forest soils around the world (Yu et al. 2017). Among all three groups and years, the 2015 saturated plots had the highest proportion of CH$_4$ emissions exceeding 0.5 mg C-CH$_4$ m$^{-2}$ h$^{-1}$ (6.5 %) and highest number of ebullition events (26.6 %). Vegetated collars had a higher median CH$_4$ flux than in plots with vegetation excluded, particularly in the saturated plots (Figure 3).





There appeared to be an increase in $CH_4$ emissions over time in the vegetated collars within the saturated plots both within the growing season and inter-annually.

The rates of $Mn^{2+}$, $Fe^{2+}$ and $NH_4^+$ ion adsorption on the PRS probes were greatest in the saturated lowland plots and lowest in the upland plots. By 2015, net adsorption of S was lower in the saturated lowland plots than in the upland plots (Figure 4). The ion with the strongest absolute temporal trend in adsorption was S (Table 2, Figure 4), with most of this trend likely a result of changes in $SO_4^{2-}$ availability (Geer and Schoenau, 1994; Li et al., 2001). In the saturated plots, mobile S decreased seasonally, annually and over the duration of the study (the greatest Spearman rank

coefficient was for 0.25 m deep S ion adsorption rates and chronological time represented as increasing values 1 through 9, Table 2). Declines in net S adsorption were also observed in the other plot groups but the greatest decline was in the saturated lowland plots, where relative to 2013 values, 2014 values were 40 % lower and 2015 values 50 % lower (Table 1 and Figure 4). Correspondingly, net adsorption rates for $Mn^{2+}$, $Fe^{2+}$ and $NH_4^+$ ions significantly increased over time at the saturated plots (Table 2).

**4.2 Spatial and temporal relationships along a moisture gradient in the lowland**

Along the moisture gradient established by a slightly sloping surface in the northeastern part of the lowland (Figure 1), $CH_4$ emissions were higher in the 4 saturated plots than in the 5 unsaturated plots (Figure 5). Methane emissions were also higher in vegetated plots than in plots with vegetation excluded. Using a LME model (Eq. 1), and controlling for sampling location as a random effect, $CH_4$ emissions were $0.34 \pm 0.11$ mg m$^{-2}$ h$^{-1}$ greater in saturated plots than

265 unsaturated and $0.17 \pm 0.06$ mg m$^{-2}$ h$^{-1}$ greater in vegetated collars than collars without vegetation over the August period (Table 3), when $CH_4$ observations were greatest, both in magnitude and proportion of emissions exceeding 0.5 mg C-$CH_4$ m$^{-2}$ h$^{-1}$ (Table 1). Temperature and soil moisture were also included in an earlier version of the LME model, but they did not have a significant effect on $CH_4$ emissions so were removed from the model.

When $CH_4$ emissions were greatest in the saturated plots, both REDOX potential and the PRS ion adsorption rates

indicated reduced soil conditions (higher $Mn^{2+}$ and $Fe^{2+}$ and lower S net adsorption; Figure 5). Using just the August data from all lowland plots the LME model indicated that the average net adsorption of $Mn^{2+}$ was $16.0 \pm 3.7$ µg 10 cm$^{-2}$ month$^{-1}$ higher in saturated conditions, and $7.1 \pm 2.3$ µg 10 m$^{-2}$ month$^{-1}$ higher for the deeper (0.2 m) probes. The net adsorption of $Fe^{2+}$ was $188 \pm 36$ µg 10 m$^{-2}$ month$^{-1}$ higher in saturated conditions but there was no significant difference at depth ($\beta_2 = 42 \pm 24$). The net adsorption of S was $588 \pm 92$ µg 10 m$^{-2}$ month$^{-1}$ lower in the saturated plots,

but there was no significant difference between the two depths ($\beta_2 = -74 \pm 63$). The median REDOX potentials were -127 and -168 mV in the saturated plots at 0.2 and 0.4 m below the surface, respectively. The LME model (Eq. 2) indicated REDOX potentials were 116 mV $\pm$ 47 (hydrogen standard) lower in saturated conditions and 85 mV $\pm$ 33 lower at a 0.4 m depth (Table 3).

The PCA's leading component related increasing net adsorption rates of $NH_4^+$ and the metals with decreasing S.

Together, the leading two components of the PCA explained 89.8 % of the variability in net ion adsorption rates for the deep probes (with 79.1 % explained by the first component) for the nine plots with REDOX potential measurements. All ions loaded relatively evenly in magnitude on the leading component (PC1; Table 4), but S was inverse to the others. Although the negative sign of the correlation coefficient between PC1 and the REDOX potential





(R = -0.24) intuitively matched what would be expected for a REDOX gradient, where more negative potentials were
285 associated with less S availability and more $Mn^{2+}$ and $Fe^{2+}$, it was not significantly different from zero (p = 0.243).
The correlation coefficient between PC1 and log transformed $CH_4$ emissions (R = 0.378) was also not significant (p
= 0.06), perhaps due to relatively few observations at the REDOX monitoring transect in 2015 (n=25). However, the
same analysis conducted for all plots' PRS data (n=163) found similar weighting on PC1 and a significant (p < 0.001)
correlation between PC1 and log-transformed methane fluxes (R = 0.31). This suggests a link between increasing $CH_4$
emissions to decreasing S and increasing $Mn^{2+}$, $Fe^{2+}$ and $NH_4^+$ availability. Some of the individual ion adsorption rates
were correlated to log transformed $CH_4$ emissions for the 9 plots along the moisture gradient in the lowland and for
the entire dataset (Table 5). At the REDOX monitoring transect in 2015, only S adsorption was negatively and
significantly correlated to $CH_4$ emissions, but when the full data set was included, $CH_4$ emissions were significantly
correlated with the following ion adsorption rates in the following order, by correlation strength: $Mn^{2+} = Fe^{2+} > NH_4^+$
$> NO_3^-$ (Table 5). Overall, the correlations were similar for shallow and deep probes (Table 5). A significant
correlation was also found between $CH_4$ fluxes and 0-20 cm integrated volumetric soil moisture (R = 0.24), whereas
no significant relationship was found for 0.02 m soil temperature (R = -0.04).

## 5 Discussion

Methane emissions typically increase with increasing saturation/rising water tables within (Moore et al., 2011) and
300 among peatlands (Turetsky et al., 2014), and also tend to increase when drained peatlands are restored by wetting
(supplementary Table S1). However, $CH_4$ emissions in the newly constructed SFW unexpectedly remained very low
over the first three years since wetting, despite abundant vegetation with aerenchymatous tissues, peat soils, and high
water tables. Over the three-year study period, the only plots that showed small but increasing trends in $CH_4$ emissions
over time were vegetated plots with standing water above the soil surface. The vegetation at these plots was primarily
*Carex aquatilis,* and *Typha latifolia*. These species have aerenchymatous tissues that enable plant-mediated transport of
$CH_4$ to the surface and have been reported in the peatland restoration literature to promote $CH_4$ emissions (Mahmood
and Strack, 2011; Wilson et al., 2009).

Median $CH_4$ fluxes from this system are 0.2 to 50 % of the values published from other studies on rewetted peatlands
(Table 1; Table S1). Instead, the fluxes in this study are similar to those reported from the other constructed wetland
in the AOSR (median $CH_4$ emissions were below 0.08 mg $m^{-2}$ $h^{-1}$) and an undisturbed saline fen (Murray et al., 2017).
Murray et al. (2017) accredited the small $CH_4$ emissions and low $CH_4$ pore water concentrations at the constructed
wetland to the supply rate of mobile S. Methanogens, $CH_4$-producing microorganisms, are obligate anaerobes and an
abundance of alternative electron acceptors such as $SO_4^{2-}$ can support microbial communities that can outcompete
methanogens. This effect has been described using the conceptual framework of the REDOX 'ladder' (for a detailed
definition see Bethke et al., 2011). Simply, a conceptual pristine aquifer will contain zones with distinct electron
acceptors for metabolic REDOX reactions as reduction potentials decrease (Lovley et al., 1994) . The sequence starts
with the reduction of oxygen until it is consumed, then oxidized nitrogen ($NO_3^-$), oxidized metals (non-mobile $MnO_2$
and $Fe(OH)_3$) and S (e.g. $SO_4^{2-}$), finally with the production of $CH_4$ through the reduction of $CO_2$ or acetate.



In incubation studies, methanogens can be outcompeted by both metal reducing bacteria (MRB) (Achtnich et al., 1995;
Miller et al., 2015) and by sulphur reducing bacteria (SRB) (Achtnich et al., 1995; Akunna et al., 1998; Gauci and
Chapman, 2006; Granberg et al., 2001; Han-Schofl et al., 2011; Kang et al., 1998; Kuivila et al., 1989; Lovley and
Klug, 1983; Peters and Conrad, 1995; Watson and Nedwell, 1998). Although the community dynamics between MRB,
SRB and methanogens is complex (Bethke et al., 2011), increased interactions among these organisms often leads to
the suppression of methanogenic activity since microbial communities are competing for $H_2$ and acetate, which
methanogenic microbes require exclusively for their metabolism. In some fen peatlands, drought conditions have been
shown to increase alternative electron acceptor abundance and suppress $CH_4$ production after rewetting (Estop-
Aragonés et al., 2013). Two lines of evidence suggest that methanogenesis was inhibited in the SFW through this
mechanism. Within the plots with REDOX probes, the largest (albeit still very small) $CH_4$ fluxes were observed
where reduction potentials at both 0.2 and 0.4 m below the surface were close to hydrogen standards of -200 mV;
potentials known to be favourable for methanogenesis (Akunna et al., 1998). At these potentials, the net adsorption of
reduced metals ($Fe^{2+}$ and $Mn^{2+}$) were greatest and mobile S, which would be largely $SO_4^{2-}$, was lowest. In those nine
plots where REDOX was observed, only S had a significant relationship to $CH_4$, but $Fe^{2+}$, $Mn^{2+}$ were also found to
have a significant correlation to $CH_4$ with the full data set. In addition, $CH_4$ emissions and ebullition events increased
over time in the saturated plots (those with standing water) throughout the lowland while S appeared to be cycling out
of the ecosystem as the presence of mobile metals ($Fe^{2+}$ and $Mn^{2+}$) increased. This suggested the soils were becoming
more reduced and perhaps lead to a decrease in SRB abundance and eased the competitive exclusion of methanogenic
organisms. This conclusion follows Christiansen et al. (2016) who also found that $Fe^{2+}$ measured with PRS probes
was highest in conditions which promoted greater $CH_4$ emissions.

Wetter soils are expected to increase the probability of mobile ions diffusing toward the PRS probes. For example,
Wood et al. (2015) found that the variability of net adsorption rates of $Mn^{2+}$, $Fe^{2+}$ and S flux was greatest when the
soil volumetric moisture content was highest in three undisturbed wetland sites from the same region as this study.
Wood et al. (2015) interpreted these results as an increase in ion availability as well as increased mobility. However,
if either a change in availability or mobility was the driving force of the observed variability in this study, all abundant
ions should increase (or decrease) as they did in the Wood et al. (2015) study. Here, mobile S fluxes decreased in the
345 saturated soils where reduced metal ions were increasing. The loading of net adsorption rates of S on PC1 was almost
equal to the other ions, but was negative (Table 4), suggesting that the leading mode of variability in ion adsorption
rates had an opposite relationship between the reduced and oxidized ions. This suggests that microbially mediated
REDOX reactions such as those by SRB and MRB are important in the changing biogeochemistry of the SFW peat
soils. Negative correlations between time and mobile S and $NO_3^-$, while positive correlations between time and
350 reduced metals and $NH_4^+$ provide additional evidence that alternative electron acceptors are being consumed over time
(Table 2). The results presented here are comparable to a study by Kreiling et al. (2015) who found an increase in
PRS-adsorbed $Mn^{2+}$ and $Fe^{2+}$ along with a decrease in mobile S with increasing flood frequency within the Mississippi
River floodplain. They too attributed these changes in ion adsorption to decreasing REDOX potentials.

Although there is evidence that the soil REDOX conditions and alternative electron acceptor abundance is varying in
time and space at the SFW as described by the REDOX 'ladder', the variability in $CH_4$ emissions were not strongly



explained by these factors. This may be due in part to the complexity associated with electron acceptor abundance and $CH_4$ production. In one study, methanogen communities were documented to become better competitors, relative to MRB, for scarce resources in Arctic tundra soils with increasing temperatures at higher REDOX potentials (Herndon et al., 2015). Granberg et al. (2001) demonstrated that vegetation, rapid temperature shifts, and N and S deposition all

360 had significant dependent effects on $CH_4$ fluxes. In that study, the effects of the interaction terms were as large as, and sometimes inverse to the effects of each variable correlated to $CH_4$ fluxes alone. Other factors might include the role of humic substances, soil heterogeneity and microsites, salinity, pH, and a breakdown of the REDOX ladder conceptual framework. For example, there is evidence that humic substances can suppress $CH_4$ production, by becoming anaerobic election acceptors (Blodau and Deppe, 2012). Microsites (<10 µm) with limited gas and water

exchange with surrounding soil pore space also may affect overall soil REDOX potential and $CH_4$ production by permitting localized electron acceptor depletion or abundance (Sey et al., 2008). Soil salinity may also have impacted the $CH_4$ emissions from this wetland constructed on top of mine tailings; in 2013, the electrical conductivity was 792 $\pm 616$ µS cm$^{-1}$ which increased to $2163 \pm 248$ µS cm$^{-1}$ in 2015 (Biagi et al., 2019). Although the effects of salinity on $CH_4$ production are not well understood, Poffenbarger et al. (2011) reviewed the literature and found $CH_4$ emissions

were suppressed only in polyhaline wetlands (18 g L$^{-1}$ or ~24 mS cm$^{-1}$) where salinity far exceeded that of the SFW. In this constructed wetland the dominant form of cation is $Ca^{2+}$ not $Na^+$ and thus, it is unknown what effect this may have on the microbes at this time. However, $Na^+$ concentrations have been increasing from $56 \pm 52$ mg L$^{-1}$ in 2013 to $130 \pm 109$ mg L$^{-1}$ in 2015 (Biagi et al., 2019), suggesting a trend towards more common $Na^+$ dominated saline environments. The assemblage of anaerobic microbes that are thermodynamically favoured in a soil also varies with

pH since the Gibbs energies of some alternative electron accepting metabolic process, but not others, vary with hydrogen ion concentration (Bethke et al., 2011; Flynn et al., 2014). Bethke et al. (2011) concluded that at neutral pH, the Gibbs energies of the major metabolic pathways of $Fe^{3+}$ reduction, $SO_4^{2-}$ reduction, and methanogenesis all converge. This contradicts the competitive exclusion concept of the REDOX 'ladder' at neutral conditions. Other reactions can also have a cascading effect on the thermodynamics of the system. For example, Kreiling et al. (2015)

found that precipitation of $Fe^{2+}$ and $H_2PO_4^-$ lead to non-linear trends in $Fe^{2+}$ ion adsorption to ion exchange resins despite increasing time in anoxic conditions because the precipitate removed waste products and maintained the system's relative abundance of oxidized iron ($Fe^{3+}$) favourable for forward reactions within Fe reducing metabolic pathways. Currently, microsite conditions are very difficult to assess at the level Bethke et al. (2011) argue are needed to explain anaerobic microbial community dynamics on a thermodynamic basis.

At the SFW, it is reasonable to assume that N and S deposition near an oil sand processing plant could influence soil biogeochemistry and the results described here. Even before wetting, high concentrations of total S and available S in the peat used to construct the SFW were similar to an undisturbed fen in Alberta as reported by Chagué-Goff et al. (1996). Such high concentrations of S occur when the groundwater that supplies the fen passes though shale deposits containing coal or oil. Because Alberta is rich in both types of deposits, fens classified as "moderate rich" or "extreme rich" in terms

of ion abundance, such as the four boreal fens described by Hartsock et al. (2016), are not uncommon in the region. High concentrations of S are also found in natural tidal wetlands where $CH_4$ emissions are typically low (Poffenbarger et al., 2011). However, due to the abrupt change in environmental conditions affecting the salvaged peat with placement and

flooding in the SFW, the mobile S appears to be currently cycling out of the anoxic regions of this system. In the future,
surface soil $SO_4^{2-}$ may be replenished by diffusion from the underlying tailings which has high concentrations of gypsum

added post-processing to promote aggregation of soft tailings (Oil Sands Wetland Working Group, 2014). However,
there is no indication that the upward vertical transport of salts from the tailing sands is occurring at a rate to offset the
current decline in mobile S fluxes (Biagi et al., 2019).

## 6 Conclusions

Carbon cycling processes in constructed boreal plains lowlands are not yet well understood as these ecosystems have

only existed for a few years. This study shows that $CH_4$ emissions are very low in the SFW, one of the first of two
boreal plains lowlands constructed in boreal northern Alberta. Methane emissions were low compared to rewetted
and restored peatlands but similar to another newly constructed wetland in the Alberta oil sand region. Changes in ion
adsorption rates on buried ion exchange resins (PRS probes) were related to decreasing REDOX potentials and were used to
support the argument that methanogen activity may be competitively suppressed by an abundance of alternative electron

acceptors. Seasonal and interannual variations in net ion adsorption rates suggest microbially mediated changes in soil
chemistry. Concurrent modest increases in $CH_4$ emissions suggest $CH_4$ emissions from the SFW are likely to increase in the
future, if the trends in the abundance of electron acceptors continue. The findings of this research indicate that the design of
the SFW promotes, at least initially, highly reduced soils in the lowlands with limited $CH_4$ production. This is significant as the
REDOX conditions necessary for long term C accumulation appear to be achieved while limiting the emissions of a potent

greenhouse gas.

## 7 Author Contributions

Humphreys' and Carey supervised Clark during this work. The experiments were designed by Clark and Humphreys,
with Clark conducting them in the field. Funding was obtained by Carry and Humphreys. This manuscript was written
by Clark, who also performed the primary investigation, with contributions from all co-authors.

## 8 Data Availability

Data will be provided on request.

## 9 Acknowledgements

This research was supported by Syncrude Canada. We thank everyone who helped with the field work, Erin Nicholls,
Kelly Biagi, Haley Spennato, Chelsea Thorne, Jessica Sara, Arthur Szybalski, Dr. Gord Drewitt as well as Dr. Mike
Treberg for building and helping maintain the gas exchange chambers. We would also like to thank the dedicated
team at the Reclamation and Closure Department of Syncrude Canada for their support.



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





## 11 Tables and Figures

**Table 1: Soil and CH₄ flux characteristics of the three location groups. The standard deviation from the mean is reported in brackets. Dominant vegetation is the most common plant type among the vegetated collars (by % cover).**

| | | Lowland – Saturated | Lowland – Unsaturated | Upland – Unsaturated |
|---|---|---|---|---|
| Number of Plots* | 2013 | 7 | 4 | 6 |
| | 2014 | 7 | 4 | 12 |
| | 2015 | 9 | 13 | 12 |
| Median CH₄ Flux (mg C-CH₄ m⁻² h⁻¹) | 2013 | 0.00 (±0.68) | 0.00 (±0.05) | 0.00 (±0.06) |
| | 2014 | 0.02 (±0.25) | 0.00 (±0.04) | -0.01 (±0.06) |
| | 2015 | 0.04 (±0.32) | 0.01 (±0.28) | 0.00 (±0.15) |
| Maximum CH₄ Flux (mg C-CH₄ m⁻² h⁻¹) | 2013 | 6.64 | 0.20 | 0.42 |
| | 2014 | 2.04 | 0.08 | 0.43 |
| | 2015 | 2.14 | 3.77 | 2.07 |
| Proportion of flux measurements with ebullition / proportion of CH₄ emissions > 0.5 mg C-CH₄ m⁻² h⁻¹ | 2013 | 10.0 %/2.1 % | 8.0 %/0.0 % | 1.6 %/0.0 % |
| | 2014 | 21.4 %/2.0 % | 14.5 %/0.0 % | 7.7 %/0.0 % |
| | 2015 | 26.6 %/6.5 % | 5.4 %/1.4 % | 6.3 %/0.7 % |
| Mean 0-0.2 m volumetric soil moisture (%, ±1 SD)¹ | 2013 | 87.0 (±0.4) | 51.3 (±20.7) | 55.1 (±34.6) |
| | 2014 | 87.0 (±0.0) | 54.2 (±12.7) | 34.5 (±21.4) |
| | 2015 | 87.0 (±0.0) | 59.2 (±9.7) | 29.3 (±20.4) |
| Mean 0.02 m Soil Temperature (ºC, ±1 SD) | 2013 | 14.4 (±7.8) | 12.2 (±9.4) | 17.2 (±8.4) |
| | 2014 | 15.9 (±9.0) | 14.0 (±7.8) | 19.0 (±7.0) |
| | 2015 | 14.0 (±7.7) | 16.4 (±7.1) | 17.0 (±6.8) |
| Mean sulphur adsorption rates at PRS probes (µg of S m⁻² month⁻¹) | 2013 | 1206 (±186) | 1324(±144) | 828 (±534) |
| | 2014 | 718 (±430) | 1321(±124) | 857 (±402) |
| | 2015 | 606.0 (±277.0) | 1240.0 (±238.5) | 907.4 (±423.2) |
| Number of chamber measurements | 2013 | 126 | 58 | 124 |
| | 2014 | 77 | 71 | 216 |
| | 2015 | 135 | 262 | 252 |
| Dominant plant types within the vegetated collars | 2013 | Sedges, grasses | Grasses, sedges, shrubs | Grasses, herbs, shrubs |
| | 2014 | Cattails, sedges, rushes | Grasses, sedges, herbs, shrubs, rushes, cattails | Grasses, herbs, shrubs |
| | 2015 | Cattails, sedges, rushes | Grasses, sedges, herbs, shrubs, rushes, cattails | Grasses, herbs, shrubs |

*As the water table declined, plots were reclassified as unsaturated in the lowland at the time of sampling. Therefore, the sum of the groups each year exceeds the number of chambers reported in the text for that year.

¹When standing water was observed, soil moisture was not measured, and the volumetric soil moisture was set to 87 %. The depth of standing water varied from 0.03 m to 0.33 m.





**Table 2: Spearman rank correlations to assess temporal trends in net ion adsorption rates for the three plot groups. Plant Root Simulator (PRS) data were categorized into chronological order (1-9 for the three burial periods over three years, 'Combined'), seasonal order regardless of year (1-3, 'Seasonal'), and annual order regardless of season (1-3, 'Annual'). Bold values indicate a significant trend at α = 0.01. The greatest temporal trend across all plot groups for a given ion and depth is shaded in black.**

|  | Ion | Lowland saturated | | | Lowland unsaturated | | | Upland | | |
|---|---|---|---|---|---|---|---|---|---|---|
|  |  | Combined | Seasonal | Annual | Combined | Seasonal | Annual | Combined | Seasonal | Annual |
| Shallow | $Mn^{2+}$ | **0.26** | **0.22** | **0.21** | **-0.21** | **-0.29** | -0.11 | **-0.35** | **-0.32** | **-0.23** |
| Deep | $Mn^{2+}$ | **0.54** | **0.17** | **0.52** | **0.49** | 0.07 | **0.50** | **-0.27** | **-0.29** | **-0.17** |
| Shallow | $Fe^{2+}$ | 0.47 | **0.48** | **0.37** | **0.26** | 0.06 | 0.29 | -0.01 | **-0.26** | 0.08 |
| Deep | $Fe^{2+}$ | **0.64** | **0.34** | **0.60** | **0.54** | **0.26** | **0.52** | 0.07 | **-0.16** | 0.10 |
| Shallow | S | **-0.64** | **-0.45** | **-0.54** | 0.03 | **-0.31** | **0.15** | -0.02 | -0.08 | 0.05 |
| Deep | S | **-0.74** | **-0.32** | **-0.71** | **-0.43** | **-0.48** | **-0.25** | -0.08 | **-0.20** | 0.03 |
| Shallow | TN | **-0.19** | -0.10 | **-0.19** | **-0.33** | **-0.19** | **-0.22** | **-0.17** | 0.04 | **-0.19** |
| Deep | TN | -0.05 | -0.09 | -0.03 | **-0.28** | -0.02 | **-0.28** | **-0.25** | -0.01 | **-0.25** |
| Shallow | $NO_3^-$ | **-0.36** | **-0.36** | **-0.26** | **-0.30** | **-0.35** | -0.10 | -0.08 | -0.06 | -0.04 |
| Deep | $NO_3^-$ | -0.14 | **-0.26** | -0.05 | **-0.35** | **-0.27** | **-0.22** | **-0.15** | -0.04 | **-0.13** |
| Shallow | $NH_4^+$ | **0.19** | **0.18** | 0.15 | -0.14 | **0.17** | **-0.16** | **-0.33** | **0.43** | **-0.54** |
| Deep | $NH_4^+$ | **0.21** | **0.25** | **0.16** | **0.21** | **0.38** | 0.08 | **-0.37** | **0.31** | **-0.50** |





**Table 3:** Results of the linear mixed effects models for CH4 emissions, PRS ions and REDOX potential (Equations 1 and 2) measured August 2015 at the nine plots along the moisture gradient in the lowland of the Sandhill Fen Watershed. Values are the mean effect ± the standard deviation. Bold values indicate parameters significantly different from zero at the $\alpha = 0.05$ level. Vegetation effects are only for the CH4 emission model (Equation 1). Plot effect is the standard deviation of the random effects from each plot.

| Response Variable | F-value | $\beta_1$ (Saturated) | $\beta_2$ (Depth/Vegetation) | $\alpha$ (Intercept) | $Var(\zeta)^{\frac{1}{2}}$ (Plot Effect) |
|---|---|---|---|---|---|
| CH4 | 0.84 | **0.34 ± 0.11** mg m$^{-2}$ h$^{-1}$ | **0.17 ± 0.06** mg m$^{-2}$ h$^{-1}$ | -0.04 ± 0.07 mg m$^{-2}$ h$^{-1}$ | 0.18 mg m$^{-2}$ h$^{-1}$ |
| Mn$^{2+}$ | 3.12 | **16.0 ± 3.7** µg 10 cm$^{-2}$ month$^{-1}$ | **7.1 ± 2.3** µg 10 cm$^{-2}$ month$^{-1}$ | 1.7 ± 2.5 µg 10 cm$^{-2}$ month$^{-1}$ | 5.5 µg 10 cm$^{-2}$ month$^{-1}$ |
| Fe$^{2+}$ | 2.99 | **188 ± 36** µg 10 cm$^{-2}$ month$^{-1}$ | 42 ± 24 µg 10 m$^{-2}$ cmonth$^{-1}$ | **57 ± 25** µg 10 cm$^{-2}$ month$^{-1}$ | 51 µg 10 cm$^{-2}$ month$^{-1}$ |
| S | 3.8 | **-588 ± 92** µg 10 cm$^{-2}$ month$^{-1}$ | -74 ± 62 µg 10 m$^{-2}$ cmonth$^{-1}$ | **1145 ± 64** µg 10 cm$^{-2}$ month$^{-1}$ | 126 µg 10 cm$^{-2}$ month$^{-1}$ |
| REDOX | 0.42 | **-116 ± 47** mV | **-85 ± 33** mV | 13 ± 32 mV | 44 mV |





**Table 4: PCA loadings and the Pearson correlation for principle components (PC) 1 and 2 and REDOX potential and log transformed CH₄ flux (p-values given in parentheses). Methane fluxes were averaged over the same one-month periods the PRS probes were buried. Bold numbers indicate significant correlations at α = 0.05, n = 25.**

| PC loadings: | | | | Pearson correlations: | | |
|---|---|---|---|---|---|---|
| Ion | PC1 | PC2 | | Predictor | PC1 | PC2 |
| $NH_4^+$ | 0.41 | 0.76 | | REDOX 0.2 m | -0.275 (0.166) | 0.254 (0.201) |
| $Mn^{2+}$ | 0.54 | 0.1 | | REDOX 0.4 m | -0.06 (0.767) | -0.025 (0.901) |
| $Fe^{2+}$ | 0.58 | -0.15 | | ln(CH₄ flux) | **0.295** (<0.001) | **0.187** (0.016) |
| S | -0.45 | -0.62 | | | | |

**Table 5: Pearson correlation coefficients between natural logarithm transformed CH₄ fluxes and Plant Root Simulator (PRS) ion exchange resin measurements. Methane fluxes were averaged over the same one-month periods the PRS probes were buried, n = 25 for transect data n= 163 for entire study. Deep probes were buried outside the collars at 0.2 m, shallow probes were buried at 0.1 m outside the collars. Significant correlations (α = 0.05) are in bold.**

| | 2015 Transect Only | | All Data | |
|---|---|---|---|---|
| | PRS Probe Location | | PRS Probe Location | |
| Ion | Deep | Shallow | Deep | Shallow |
| $NO_3$ | -0.21 | -0.37 | **-0.14** | **-0.16** |
| $NH_4^+$ | 0.36 | 0.13 | **0.23** | **0.15** |
| $Mn^{2+}$ | 0.34 | 0.17 | **0.31** | **0.37** |
| $Fe^{2+}$ | 0.31 | 0.40 | **0.31** | **0.38** |
| S | **-0.47** | -0.36 | -0.06 | 0.12 |



**Figure 1: Map of the Sandhill Fen Watershed. Standing water was limited to the lowland region only.**

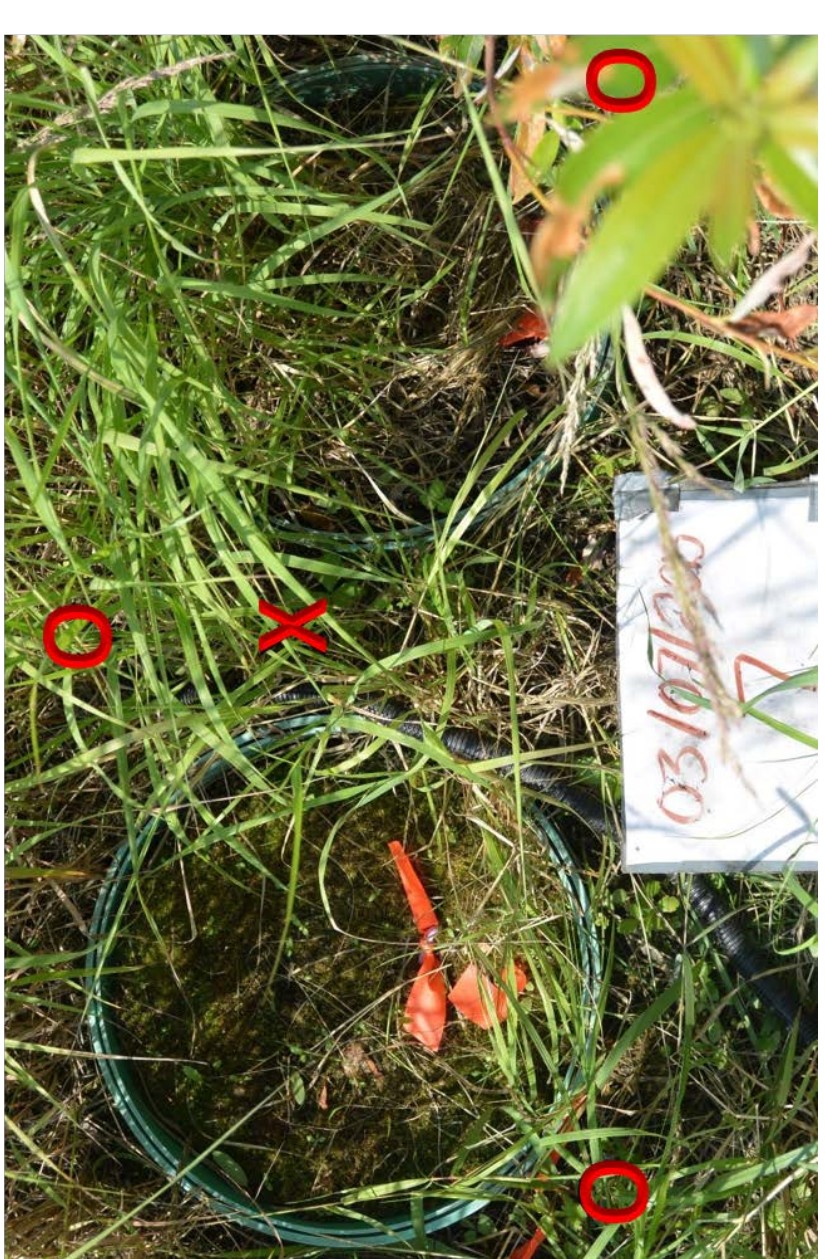

**Figure 2: Example of a plot with a collar pair in the wetland area illustrating a collar excluding vegetation (on the left) and a vegetated collar (right), surface PRS probes (orange and purple tabs with orange flagging tape, locations outside the collar are marked with circles). The permanent soil thermocouple profile post is marked by an 'X'. This plot is from the lowland unsaturated category (bottom of the eastern most boardwalk, Figure 1) and the photograph was taken July 3rd 2015.**



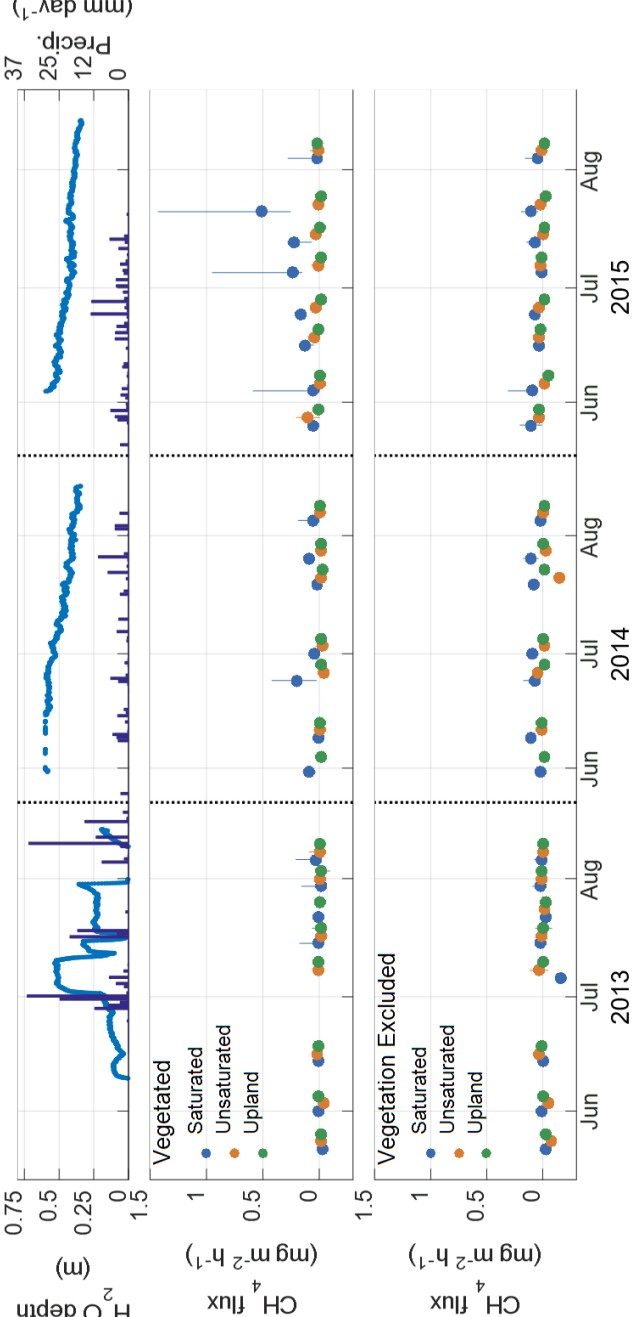

**Figure 3:** Water table, precipitation and median CH$_4$ fluxes by sampling date. Water table depth and precipitation (top panel) were measured at the central meteorological station (Figure 1). Circles indicate median CH$_4$ flux on the day of sampling and lines indicate the 25$^{th}$ to 75$^{th}$ percentiles for collars with vegetation (middle panel) and collars maintained free of vegetation (bottom panel). An offset between vegetation groups has been added to the x-axis to reduce overlapping points. All groups were sampled on the same day, roughly once a week. Tick marks on the x-axis indicate first day of the labelled month. If the upper 75$^{th}$ percentile exceeded the range of the y axis, the value is printed on the plot.



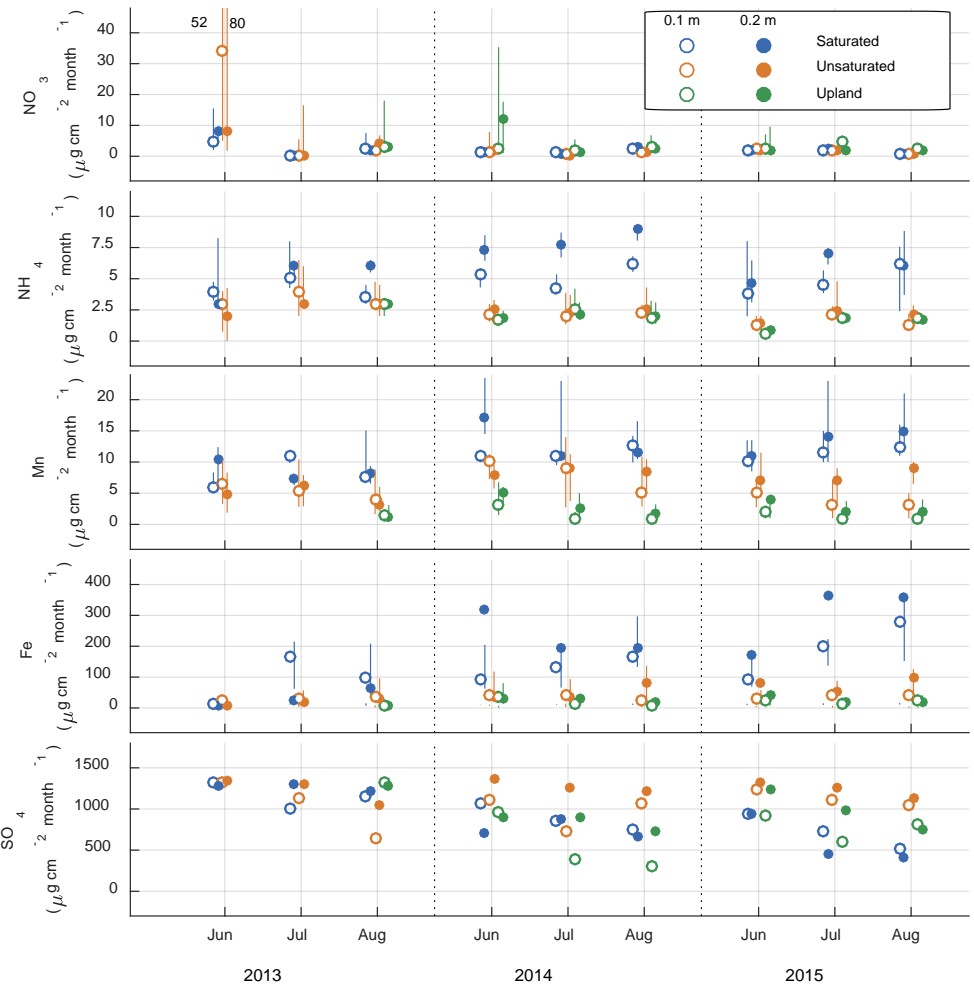

**Figure 4: Ion availability as measured using Plant Root Simulator ion exchange resins over the three monthly burial periods for three years expressed in units of μg 10 cm-2 month-1 at two depths (0.1 m and 0.2 m). Units are a function of time the probes were buried for. Metal ions were measured in their reduced forms and mobile S is the oxidized form of sulphur. An offset has been added between groups on the x-axis to reduce overlapping points.**



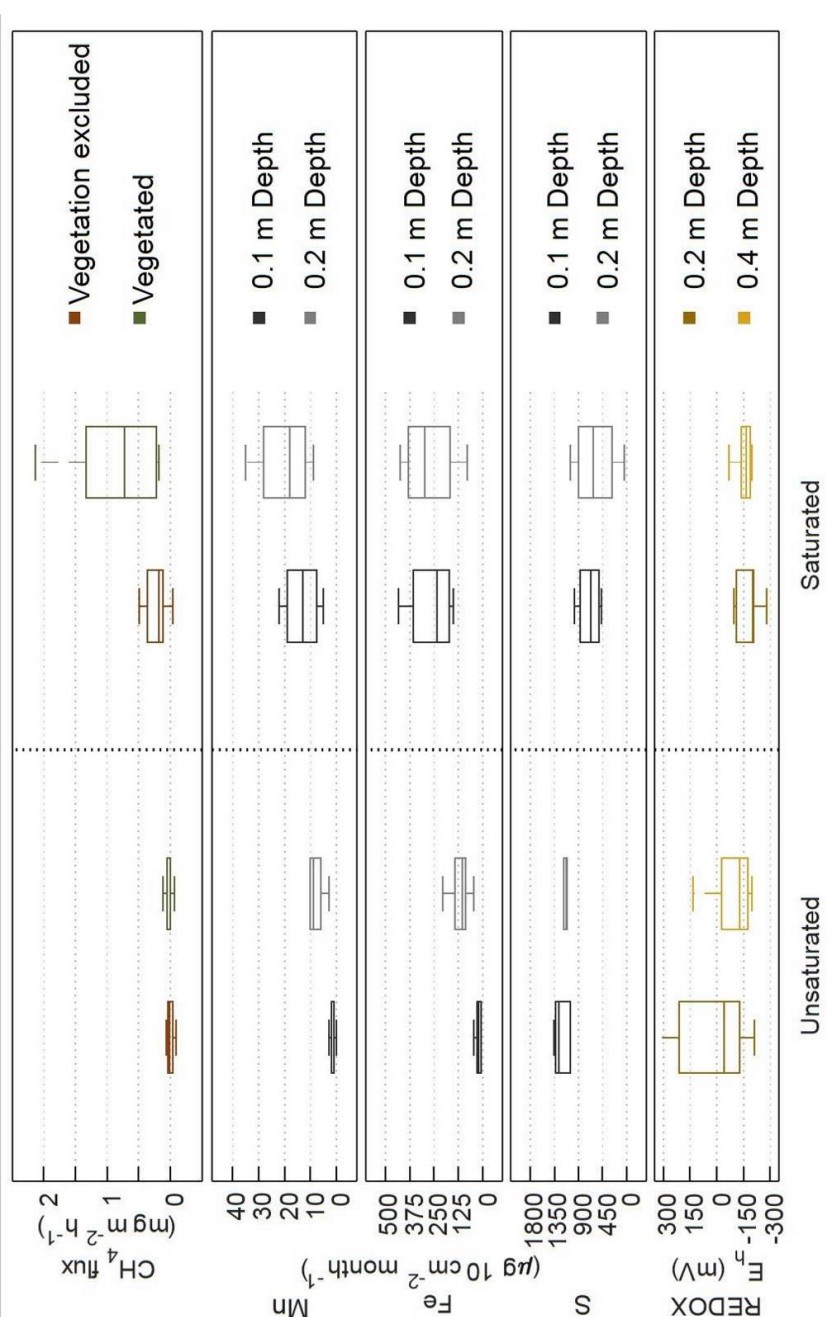

**Figure 5: CH4 fluxes (top panel), net rates of ion adsorption measured with PRS ion exchange resins (middle panels) and the REDOX potential (bottom panel) measured every 15 min at 0.2 m and 0.4 m over the burial period in the lowland moisture gradient plots August 2015. Boxplots represent the interquartile range, with whiskers as 95 % and lines representing the median for the nine plots (n = 4 for the saturated category where there was standing water, n = 5 for the unsaturated category, where mean 0-0.20 m volumetric water content was 57 %)**