# Peer review of "Low methane emissions from a boreal wetland constructed on oil sand mine tailings"

_Biogeosciences, 2019_

## Referee Comment (RC1) · Anonymous Referee #1 · 11 Sep 2019

General comments

This paper presents results from a three year study of methane (CH4) emissions across a constructed watershed in the Athabasca Oil Sands region. The watershed contains both upland and lowland areas and was constructed on a former mine site. The authors measured low rates of CH4 emission, even at lowland plots that were frequently flooded. However, at these saturated sites, CH4 emission increased over the three years of the study. This was linked with a shift in availability of inorganic terminal electron acceptors as measured by resin probes. Therefore, the authors conclude that CH4 emission at the site is limited by the competition with metal and sulphate reducing bacteria with continued reduction of electron acceptors over time leading to the observed increase in CH4 emissions.

[Figure]

As there are few studies of biogeochemistry in constructed landscapes, particularly peatlands in the oil sands region of northern Alberta, Canada, this study provides new insight to carbon cycling and greenhouse gas exchange in the region. The work was carefully conducted and the results are clearly presented. The discussion is clear and concise. My only substantive comments is the use of the word ebullition and clarity in the methods on how this was defined. I have highlighted a few questions about this in specific comments, but overall, I was unclear when it came to the results, exactly what "ebullition" was referring to and how this was determined. This should be solved with some clarification in the methods.

Specific comments

Abstract – often ebullition is reported as a % of emission and even though you state that it is ratio to total number of sampling events, might be clearer to reword here as "Ebullition events occurred in 10% of measurements in 2013, increasing to 21 and 27% of measurement in 2014 and 2015, respectively at the plots with saturated soils"

I suggest rewording the results for the PCA to something like: "Using principal component (PC) analysis, methane fluxes significantly correlated to PC1 that was associated with ammonium, iron, manganese and sulphur availability". But can you also specify the direction of the correlation?

You should clarify – "alternative inorganic electron acceptors"

Line 21: 20 is really a low GWP for CH4. The Fifth Assessment Report (IPCC 2013) gives 28-34 depending on whether climate-carbon feedbacks are considered. This would be a more appropriate value to cite.

Lines 50-61: This is all fine, but it would also be good to have some overview of how rewetted peatlands compare to undisturbed sites. This sort of makes it sounds like rewetting is creating a source of CH4 on the landscape (because of course flux is higher than from drained sites), but in reality it is likely just returning the flux back to

what it would have been prior to the drainage disturbance.

Line 83: I suggest that you incorporate the note (known to contain salts and naphthenic acids) up with the definition of process-affected waters on Lines 80-81. This will help to avoid interjecting many sets of brackets.

Lines 125-126: So did inclusion in the saturated/unsaturated groups change over time, i.e., if a plot was saturated one week and then not the other, did it move between groups? What implications would this have on the analysis?

Line 128: Can you also state the horizontal dimensions of the collar here?

Line 133: So why not use taller chambers or bigger collars?

Line 166: How did you define "spike", especially at the start of the measurement? Was there a concentration threshold? A change in slope or first difference threshold?

Line 207: By saying the leading two principal components were correlated, it sounds like you are already stating a result. I know what you mean, but I think it should be reworded. For example, you could say "We used Pearson correlation to assess relationships between the first two principle components. . ."

Line 242: So these ebullition events – is this based on the "spikes" discussed in the method, or something else, based on the magnitude of flux.

Line 334: Is it cycling out of the system or out of solution? Do you think there is now a large reduced sulphur pool that could be remobilized if the site becomes drier? Or is the S lost as a gas?

Line 364: And more recently, the solid organic matter itself has been shown to be an important electron acceptor in peat soils (Gao et al. 2019).

Lines 379-383: This is a really long sentence. I suggest breaking it into two sentence to improve readability.

Line 397: So, what implications might this have on future CH4 emissions?

Technical corrections: Line 5: Typo in reference – should be Environmental

Lines 15-16: I don't think page numbers are needed in the referencing for the journal.

Line 20: its not it's

Line 159: Should be linear not liner.

References Gao C, Sander M, Agethen S, Knorr K-H. 2019. Electron accepting capacity of dissolved and particulate organic matter control CO2 and CH4 formation in peat soils. Geochemica et Cosmochimica Acta, 245, 266-277.

IPCC. 2013. Climate Change 2013: The Physical Science Basis. Contribution of Working Group I to the Fifth Assessment Report of the Intergovernmental Panel on Climate Change [Stocker TF, Qin D, Plattner G-K, Tignor M, Allen SK, Boschung J, Nauels A, Xia Y, Bex V, Midgley PM (eds.)]. Cambridge University Press, Cambridge, United Kingdom and New York, NY, USA, 1535 pp.

————————————————————

---

## Editor Comment (EC1) · Tina Treude (Editor) · 14 Nov 2019

Dear Dr. Clark and Co-Workers, since the first review of your manuscript was very positive (minor changes) and the waiting period to find a second reviewer has unfortunately been very long, I decided to work with this one review. The discussion period can be closed. Tina Treude

---

## Author Comment (AC1) · 29 Nov 2019

The following response is structured in the order upon which the comments were made by RC1.

Specific comments:

We agree with the reviewer, their wording clarifies the percentage as a proportion of samples, not total emission. We made this first suggested correction verbatim.

We rewrote the sentence to reflect the clarifications suggested by the reviewer. The sentence now reads "Using principal component (PC) analysis, methane fluxes had a significant positive correlation to the leading PC which was associated with increasing ammonium, iron, and manganese availability and deceasing sulphur availability (r =

0.31, p < 0.001). "

We added inorganic to "alternative inorganic electron acceptors" as suggested for clarity.

On line 21 we updated reference and value to the fifth assessment report standards as suggested by RC1.

Comments for lines 50-61: The literature is limited in comparison to undisturbed sites. To highlight how much methane undisturbed sites produce, we changed a sentence to show their impact on the global methane budget. In the prior paragraph, where discussing wetland emissions, we made the following changes: "Methane emissions from wetlands are highly variable in space and time (Moore et al., 1998), but are significant sources of atmospheric methane. Globally wetlands represent 32% of the total sources of atmospheric methane (IPCC 2013)." We also added the following sentence at the end of the paragraph in question: "The few studies which compare emissions from rewetted and undisturbed wetlands (supplementary Table S1) show a wide range of results with rewetted wetland emissions <1% (Juottonen et al., 2012), 19% (Beetz et al., 2013), 43% (Urbanová et al., 2012), and 127% (Christen et al., 2016) of the emissions observed in undisturbed wetlands."

The note on line 83 was moved as suggested.

RC1 questioned the changes in grouping as outlined on lines 125-126. Five of the sites did change at various times. In 2013, when the water table was lower and heavily managed four sites moved between the saturated and unsaturated groups, but methane was almost absent in that year. In 2014 and 2015, the switch occurred in early spring (first week of June) before methane production increased as discussed. So we believe that this categorization had little impact on the results of this paper. More recent work by Dale Vitt and Jeremey Hartsock in the Sandhill Fen (personal communication) is demonstrating that PRS probes tend to reflect the ions towards the end of their burial period. This makes us confidant that the early spring change of group likely did not

impact the ion results as they likely were biased towards ion fluxes at the end of the burial period when the collars were stable in their grouping. We added the statement, "Only 5 lowland plots switched categories in 2013 and early spring 2014 and 2015, all periods when CH4 emissions were uniformly low with no discernible differences among the groups."

Moved the comment "with a surface area of 0.07 m2" up to address bring all size discussion to line 128, as suggested.

In response to why we didn't use a larger chamber: It wasn't a regular occurrence and we didn't have the capacity on hand to build new chambers, so it was either trim or lose the data for that whole season. Since it was only relevant at a few sites, we decided to trim the vegetation and save what data we could.

Clarified this sentence on line 166 to be "These anomalies included isolated large decreases in concentration or a return to ambient concentration or isolated or unsustained large increases in concentration and represented fluxes from a leaking chamber or an ebullition event (Tokida et al., 2007)."

Changed the language around correlating PC's to "Using Pearson correlation, the relationship between the leading two principal components and 0.2 m REDOX measurements were assessed. Pearson correlation was also used to assess the relationship between the leading two principal components and the logarithm transformed burial period averaged CH4 flux (transformed to account for skew)."

We agree the sentence on line 242 was unclear. Since fluxes were very small in general, we intend to highlight occasions were relatively more CH4 is emitted including through ebullition. We have revised the description as, "This included occasions with ebullition or when fluxes were greater than 0.5 mg C-CH4 m-2 h-1, which is equivalent to ten times the maximum CH4 uptake rate observed at this site and is in the upper range of average uptake rates observed in grassland and forest soils around the world (Yu et al. 2017)."

RC1 asked about the fate of S in the system. That was a good question. I don't think we can say. The sentence has been changed to "...mobile S appeared to be declining in abundance as the presence of mobile metals (Fe and Mn) increased." so as not to over-interpret the results.

Thank you for bringing the Gao et al. 2019 paper to our attention. Very interesting, we added the comment as suggested.

Good idea, the long sentence was changed into the two following two sentences: "For example, Kreiling et al. (2015) found that precipitation of $Fe_{2+}$ and $H_2PO_{4-}$ lead to non-linear trends in $Fe_{2+}$ ion adsorption to ion exchange resins despite increasing time in anoxic conditions. Kreiling et al. (2015) demonstrated that the precipitate removed waste products and maintained the system's relative abundance of oxidized iron ($Fe_{3+}$), thereby maintaining favourable conditions for forward reactions within Fe reducing metabolic pathways."

Add the following sentence to speculate on future emissions as requested by RC1. "Therefore, without any other processes limiting production, $CH_4$ emission may increase in the future.

Technical Corrections Thank you RC1 for catching these. We made all the technical corrections.

---

## Author Comment (AC2) · 29 Nov 2019

Thank you for allowing us to make the suggested changes by RC1.

---

## Author Response (AR1)

Anonymous Referee #1 Received and published:

11 September 2019

**General comments**

This paper presents results from a three year study of methane (CH4) emissions across a constructed watershed in the Athabasca Oil Sands region. The watershed contains both upland and lowland areas and was constructed on a former mine site. The authors measured low rates of CH4 emission, even at lowland plots that were frequently flooded. However, at these saturated sites, CH4 emission increased over the three years of the study. This was linked with a shift in availability of inorganic terminal electron acceptors as measured by resin probes. Therefore, the authors conclude that CH4 emission at the site is limited by the competition with metal and sulphate reducing bacteria with continued reduction of electron acceptors over time leading to the observed increase in CH4 emissions.

As there are few studies of biogeochemistry in constructed landscapes, particularly peatlands in the oil sands region of northern Alberta, Canada, this study provides new insight to carbon cycling and greenhouse gas exchange in the region. The work was carefully conducted and the results are clearly presented. The discussion is clear and concise. My only substantive comments is the use of the word ebullition and clarity in the methods on how this was defined. I have highlighted a few questions about this in specific comments, but overall, I was unclear when it came to the results, exactly what "ebullition" was referring to and how this was determined. This should be solved with some clarification in the methods.

**Specific comments**

Abstract – often ebullition is reported as a % of emission and even though you state that it is ratio to total number of sampling events, might be clearer to reword here as "Ebullition events occurred in 10% of measurements in 2013, increasing to 21 and 27% of measurement in 2014 and 2015, respectively at the plots with saturated soils"

We agree with the reviewer, their wording clarifies the percentage as a proportion of samples, not total emission. We made this correction verbatim.

I suggest rewording the results for the PCA to something like: "Using principal component (PC) analysis, methane fluxes significantly correlated to PC1 that was associated with ammonium, iron, manganese and sulphur availability". But can you also specify the direction of the correlation?

We rewrote the sentence to reflect the clarifications suggested by the reviewer.  The sentence now reads "Using principal component (PC) analysis, methane fluxes had a significant positive correlation to the leading PC which was associated with increasing ammonium, iron, and manganese availability and deceasing sulphur availability (r = 0.31, p < 0.001). "

You should clarify – "alternative inorganic electron acceptors"

Added inorganic.

Line 21: 20 is really a low GWP for CH4. The Fifth Assessment Report (IPCC 2013) gives 28-34 depending on whether climate-carbon feedbacks are considered. This would be a more appropriate value to cite.

Updated reference and value.

Lines 50-61: This is all fine, but it would also be good to have some overview of how rewetted peatlands compare to undisturbed sites. This sort of makes it sounds like rewetting is creating a source of CH4 on the landscape (because of course flux is higher than from drained sites), but in reality it is likely just returning the flux back to what it would have been prior to the drainage disturbance.

The literature is limited in comparison to undisturbed sites.  To highlight how much methane undisturbed sites produce, we changed a sentence to show their impact on the global methane budget. In the prior paragraph, where discussing wetland emissions, the following changes were made (italics represent the changes): "Methane emissions from wetlands are highly variable in space and time (Moore et al., 1998), *but are significant sources of atmospheric methane.  Globally wetlands represent 32% of the total sources of atmospheric methane (IPCC 2013).*"  We also added the following sentence at the end of the paragraph in question: "The few studies which compare emissions from rewetted and undisturbed wetlands (supplementary Table S1) show a wide range of results with rewetted wetland emissions <1% (Juottonen et al., 2012),  19% (Beetz et al., 2013), 43% (Urbanová et al., 2012), and 127% (Christen et al., 2016) of the emissions observed in undisturbed wetlands."

Line 83: I suggest that you incorporate the note (known to contain salts and naphthenic acids) up with the definition of process-affected waters on Lines 80-81. This will help to avoid interjecting many sets of brackets.

The note was moved as suggested.

Lines 125-126: So did inclusion in the saturated/unsaturated groups change over time, i.e., if a plot was saturated one week and then not the other, did it move between groups? What implications would this have on the analysis?

Five sites did change at various times.  In 2013, when the water table was lower and heavily managed four sites moved between the saturated and unsaturated groups, but methane was almost absent in that year.  In 2014 and 2015, the switch occurred in early spring (first week of June) before methane production increased as discussed.  So we believe that this categorization had little impact on the results of this paper. More recent work by Dale Vitt and Jeremey Hartsock in the Sandhill Fen (personal communication) is demonstrating that PRS probes tend to reflect the ions towards the end of their burial period.  This makes us confidant that the early spring change of group likely did not impact the ion results as they likely were biased towards ion fluxes at the end of the burial period when the collars were stable in their grouping. We added the statement, "Only 5 lowland plots switched categories in

2013 and early spring 2014 and 2015, all periods when $CH_4$ emissions were uniformly low with no discernible differences among the groups."

Line 128: Can you also state the horizontal dimensions of the collar here?

Added in "with a surface area of 0.07 $m^2$"

Line 133: So why not use taller chambers or bigger collars?

It wasn't a regular occurrence and we didn't have the capacity on hand to build new chambers, so it was either trim or lose the data for that whole season. Since it was only relevant at a few sites, we decided to trim the vegetation and save what data we could.

Line 166: How did you define "spike", especially at the start of the measurement? Was there a concentration threshold? A change in slope or first difference threshold?

Clarified this sentence which is now written as "These anomalies included isolated large decreases in concentration or a return to ambient concentration or isolated or unsustained large increases in concentration and represented fluxes from a leaking chamber or an ebullition event (Tokida et al., 2007)."

Line 207: By saying the leading two principal components were correlated, it sounds like you are already stating a result. I know what you mean, but I think it should be reworded. For example, you could say "We used Pearson correlation to assess relationships between the first two principle components. . ."

Changed to "Using Pearson correlation, the relationship between the leading two principal components and 0.2 m REDOX measurements were assessed. Pearson correlation was also used to assess the relationship between the leading two principal components and the logarithm transformed burial period averaged CH4 flux (transformed to account for skew)."

Line 242: So these ebullition events – is this based on the "spikes" discussed in the method, or something else, based on the magnitude of flux.

The sentence was unclear. Since fluxes were very small in general, we intend to highlight occasions were relatively more CH4 is emitted including through ebullition. We have revised the description as, "This included occasions with ebullition or when fluxes were greater than 0.5 mg C-$CH_4$ $m^{-2}$ $h^{-1}$, which is equivalent to ten times the maximum $CH_4$ uptake rate observed at this site and is in the upper range of average uptake rates observed in grassland and forest soils around the world (Yu et al. 2017)."

Line 334: Is it cycling out of the system or out of solution? Do you think there is now a large reduced sulphur pool that could be remobilized if the site becomes drier? Or is the S lost as a gas?

Good question. I don't think we can say. The sentence has been changed to "…mobile S appeared to be declining in abundance as the presence of mobile metals (Fe and Mn) increased."

Line 364: And more recently, the solid organic matter itself has been shown to be an important electron acceptor in peat soils (Gao et al. 2019).

Thank you for bringing this paper to our attention. Very interesting, we added the comment as suggested.

Lines 379-383: This is a really long sentence. I suggest breaking it into two sentence to improve readability.

Good idea, the sentence was changed into the two following sentences: "For example, Kreiling et al. (2015) found that precipitation of $Fe^{2+}$ and $H_2PO_4^-$ lead to non-linear trends in $Fe^{2+}$ ion adsorption to ion exchange resins despite increasing time in anoxic conditions. Kreiling et al. (2015) demonstrated that the precipitate removed waste products and maintained the system's relative abundance of oxidized iron ($Fe^{3+}$), thereby maintaining favourable conditions for forward reactions within Fe reducing metabolic pathways."

Line 397: So, what implications might this have on future CH4 emissions?

Add the following sentence. "Therefore, without any other processes limiting production, $CH_4$ emission may increase in the future.

**Technical corrections:**

Line 5: Typo in reference – should be Environmental
Lines 15-16: I don't think page numbers are needed in the referencing for the journal.
Line 20: its not it's
Line 159: Should be linear not liner.

Thank you to the reviewer for catching these mistakes. We fixed them all.

**References**

[revised manuscript text omitted]
_4$ | 0.84 | **$0.34 \pm 0.11$** mg m$^{-2}$ h$^{-1}$ | **$0.17 \pm 0.06$** mg m$^{-2}$ h$^{-1}$ | $-0.04 \pm 0.07$ mg m$^{-2}$ h$^{-1}$ | $0.18$ mg m$^{-2}$ h$^{-1}$ |
| $Mn^{2+}$ | 3.12 | **$16.0 \pm 3.7$** µg 10 cm$^{-2}$ month$^{-1}$ | **$7.1 \pm 2.3$** µg 10 cm$^{-2}$ month$^{-1}$ | $1.7 \pm 2.5$ µg 10 cm$^{-2}$ month$^{-1}$ | $5.5$ µg 10 cm$^{-2}$ month$^{-1}$ |
| $Fe^{2+}$ | 2.99 | **$188 \pm 36$** µg 10 cm$^{-2}$ month$^{-1}$ | $42 \pm 24$ µg 10 m$^{-2}$ cmonth$^{-1}$ | **$57 \pm 25$** µg 10 cm$^{-2}$ month$^{-1}$ | $51$ µg 10 cm$^{-2}$ month$^{-1}$ |
| S | 3.8 | **$-588 \pm 92$** µg 10 cm$^{-2}$ month$^{-1}$ | $-74 \pm 62$ µg 10 m$^{-2}$ cmonth$^{-1}$ | **$1145 \pm 64$** µg 10 cm$^{-2}$ month$^{-1}$ | $126$ µg 10 cm$^{-2}$ month$^{-1}$ |
| REDOX | 0.42 | **$-116 \pm 47$** mV | **$-85 \pm 33$** mV | $13 \pm 32$ mV | $44$ mV |

**Table 4: PCA loadings and the Pearson correlation for principle components (PC) 1 and 2 and REDOX potential and log transformed $CH_4$ flux (p-values given in parentheses). Methane fluxes were averaged over the same one-month periods the PRS probes were buried. Bold numbers indicate significant correlations at $\alpha = 0.05$, n = 25.**

| PC loadings: | | | | Pearson correlations: | | |
|---|---|---|---|---|---|---|
| Ion | PC1 | PC2 | | Predictor | PC1 | PC2 |
| $NH_4^+$ | 0.41 | 0.76 | | REDOX 0.2 m | -0.275 (0.166) | 0.254 (0.201) |
| $Mn^{2+}$ | 0.54 | 0.1 | | REDOX 0.4 m | -0.06 (0.767) | -0.025 (0.901) |
| $Fe^{2+}$ | 0.58 | -0.15 | | ln($CH_4$ flux) | **0.295** (<0.001) | **0.187** (0.016) |
| S | -0.45 | -0.62 | | | | |

**Table 5: Pearson correlation coefficients between natural logarithm transformed $CH_4$ fluxes and Plant Root Simulator (PRS) ion exchange resin measurements. Methane fluxes were averaged over the same one-month periods the PRS probes were buried, n = 25 for transect data n= 163 for entire study. Deep probes were buried outside the collars at 0.2 m, shallow probes were buried at 0.1 m outside the collars. Significant correlations ($\alpha = 0.05$) are in bold.**

| | 2015 Transect Only | | All Data | |
|---|---|---|---|---|
| | PRS Probe Location | | PRS Probe Location | |
| Ion | Deep | Shallow | Deep | Shallow |
| $NO_3$ | -0.21 | -0.37 | **-0.14** | **-0.16** |
| $NH_4^+$ | 0.36 | 0.13 | **0.23** | **0.15** |
| $Mn^{2+}$ | 0.34 | 0.17 | **0.31** | **0.37** |
| $Fe^{2+}$ | 0.31 | 0.40 | **0.31** | **0.38** |
| S | **-0.47** | -0.36 | -0.06 | 0.12 |

[Figure]

**Figure 1: Map of the Sandhill Fen Watershed. Standing water was limited to the lowland region only.**

[Figure]

**Figure 2: Example of a plot with a collar pair in the wetland area illustrating a collar excluding vegetation (on the left) and a vegetated collar (right), surface PRS probes (orange and purple tabs with orange flagging tape, locations outside the collar are marked with circles). The permanent soil thermocouple profile post is marked by an 'X'. This plot is from the lowland unsaturated category (bottom of the eastern most boardwalk, Figure 1) and the photograph was taken July 3rd 2015.**

[Figure]

**Figure 3: Water table, precipitation and median CH4 fluxes by sampling date.** Water table depth and precipitation (top panel) were measured at the central meteorological station (Figure 1). Circles indicate median CH4 flux on the day of sampling and lines indicate the 25th to 75th percentiles for collars with vegetation (middle panel) and collars maintained free of vegetation (bottom panel). An offset between vegetation groups has been added to the x-axis to reduce overlapping points. All groups were sampled on the same day, roughly once a week. Tick marks on the x-axis indicate first day of the labelled month. If the upper 75th percentile exceeded the range of the y axis, the value is printed on the plot.

[Figure]

**Figure 4: Ion availability as measured using Plant Root Simulator ion exchange resins over the three monthly burial periods for three years expressed in units of µg 10 cm-2 month-1 at two depths (0.1 m and 0.2 m). Units are a function of time the probes were buried for. Metal ions were measured in their reduced forms and mobile S is the oxidized form of sulphur. An offset has been added between groups on the x-axis to reduce overlapping points.**

[Figure]

Figure 5: CH4 fluxes (top panel), net rates of ion adsorption measured with PRS ion exchange resins (middle panels) and the REDOX potential (bottom panel) measured every 15 min at 0.2 m and 0.4 m over the burial period in the lowland moisture gradient plots August 2015. Boxplots represent the interquartile range, with whiskers as 95 % and lines representing the median for the nine plots (n = 4 for the saturated category where there was standing water, n = 5 for the unsaturated category, where mean 0-0.20 m volumetric water content was 57 %)